# Role of Lung Ultrasound in the Management of Patients with Suspected SARS-CoV-2 Infection in the Emergency Department

**DOI:** 10.3390/jcm11082067

**Published:** 2022-04-07

**Authors:** Andrea Boccatonda, Alice Grignaschi, Antonella Maria Grazia Lanotte, Giulio Cocco, Gianpaolo Vidili, Fabrizio Giostra, Cosima Schiavone

**Affiliations:** 1Emergency Department, Istituto di Ricovero e Cura a Carattere Scientifico (IRCCS) Azienda Ospedaliero Universitaria di Bologna, 40138 Bologna, Italy; alice.grignaschi@aosp.bo.it (A.G.); antonella.lanotte@aosp.bo.it (A.M.G.L.); fabrizio.giostra@aosp.bo.it (F.G.); 2Internal Medicine, G. d’Annunzio University, 66100 Chieti, Italy; cocco.giulio@gmail.com (G.C.); cosima.schiavone@gmail.com (C.S.); 3Department of Medical, Surgical and Experimental Sciences, University of Sassari, 07100 Sassari, Italy; gianpaolovidili@uniss.it

**Keywords:** lung, ultrasound, COVID-19, SARS-CoV-2, infection, pneumonia

## Abstract

Background: The lung ultrasound (LUS) score has been proposed as an optimal scheme for the ultrasound study of patients with suspected/confirmed COVID-19 pneumonia. The aims of our study were to evaluate the use of lung ultrasound as a diagnostic tool for diagnosing SARS-CoV-2 pneumonia, to examine the validity of the LUS score for the diagnosis of COVID-19 pneumonia, and to correlate this score with hospitalization rate and 30-day mortality. Materials and Methods: A retrospective analysis was performed on 1460 patients who were referred to the General Emergency Department of the S. Orsola-Malpighi Hospital from April 2020 to May 2020 for symptoms suspected to indicate SARS-CoV-2 infection. The ultrasound examination was based on a common execution scheme called the LUS score, as previously described. Results and Conclusions: The LUS score was found to correlate with the degree of clinical severity and respiratory failure (paO_2_/FiO_2_ ratio and the alveolar–arterial gradient increase than expected for age). It was shown that COVID-19 patients with an LUS score of >7 require the use of oxygen support, and a value of >10 is associated with an increased risk of oro-tracheal intubation. The LUS score was found to present higher values in hospitalized patients, increasing according to the degree of care intensity. Patients who died from COVID-19 were characterized by a mean LUS score of 11 at presentation to the emergency department. An LUS score of >7.5 was found to indicate a sensitivity of 83% and a specificity of 89% for 30-day mortality in COVID-19 patients. The use of LUS seems to be an optimal first level method for pneumonia detection and risk stratification in patients with suspected SARS-CoV-2 infection.

## 1. Introduction

Lung ultrasound has significantly increased in importance following the emergence of the SARS-CoV-2 pandemic [1,2,3,4,5]. An exponential increase in published works has been observed from the first case reports published since the beginning of 2020 [6,7,8,9,10,11]. Lung ultrasound’s growing use is due to its simplicity, reproducibility, and relevance in the clinical setting [1,2,3,4,5]. The use of lung ultrasound seems to be of the utmost importance as a first level/screening method at the time of the first evaluation of a patient suspected for SARS-CoV-2 infection and secondly as a serial monitoring of the evolution of the disease, especially in patients who cannot be easily mobilized (e.g., intubated and ventilated patients) [1,2,3,4,5].

Several authors have suggested specific signs and scores for the study and staging of COVID-19 pneumonia, though none have reported a globally accepted scheme [2,3,4,11]. The lung ultrasound (LUS) score is a score that was validated in the pre-COVID era in the intensive care setting to monitor ventilated patients [12]; that score has been proposed by several authors as an optimal model/scheme for the ultrasound evaluation of patients with suspected/confirmed COVID-19 pneumonia [2,3,4,11].

The aims of our study were to evaluate the use of lung ultrasound as a diagnostic tool for SARS-CoV-2 pneumonia, to examine the validity of a pulmonary ultrasound score called the “LUS score” for the diagnosis of COVID-19 pneumonia, and to correlate this score with hospitalization rate and in-hospital mortality.

## 2. Materials and Methods

### 2.1. Study Design

A retrospective analysis of clinical and ultrasound data was performed. Informed consent was obtained from all patients enrolled in the study. The data were obtained from the medical records and the information system. In addition to the ordinary anamnestic and clinical evaluation, lung ultrasound and blood gas analyses were often performed as a first level method for diagnosing COVID-19 pneumonia. Subsequently, the patients were subjected to further tests, with different priorities according to the severity of the disease. The study protocol conformed to the ethical guidelines of the 1975 Helsinki Declaration, as reflected in a priori approval by the institution’s human research committee. This study was also approved by the local Ethics Committee (551/2020/Oss/AOUBo/Comitato Etico Indipendente di Area Vasta Emilia Centro (CE-AVEC)).

### 2.2. Study Patients

A retrospective analysis was performed on all patients who were referred to the General Emergency Department of the S. Orsola-Malpighi Hospital from April 2020 to May 2020 for symptoms suspected to indicate SARS-CoV-2 infection. All patients were over 18 years of age. The data analysis was performed subsequently by dividing the patients according to the outcome of the nasopharyngeal swab into COVID-19-positive and COVID-19-negative patients. Patients with a definitive SARS-CoV-2-positive result and in whom a lung ultrasound was performed were taken into consideration for the subsequent statistical analysis; patients who were not subjected to those tests were excluded from the final statistical analysis.

### 2.3. Lung Ultrasound Technique

Lung ultrasounds were performed by a pool of physicians with certified experience with the method. The ultrasound system used was an Esaote Mylab 7. The convex probe was mainly used, and the linear probe was occasionally used for the detailed evaluation of the pleural line. The ultrasound examination was based on a common execution scheme with shared ultrasound semeiotics [12] (Figure 1). The examination protocol was based on the scan of 6 lung fields per hemithorax (2 anterior, 2 lateral, and 2 posterior) for a total of 12 lung fields. The evaluation of the ultrasound signs was based on a scheme previously validated in the intensive care setting [12]; in particular, a score was assigned for each examined lung field (normal = 0; non-converging B lines = 1; confluent B lines or white lung = 2; consolidation = 3). The sum of the scores of the 12 lung fields examined provided a final result defined as the LUS score. The LUS score therefore ranged from 0 to 36. The patient was preferably examined in the sitting position; in cases of forced supine position, posterior scans were performed by rotating the patient onto their side.

## 3. Statistical Analysis

Continuous variables are expressed as mean ± standard deviation. The Mann–Whitney U test for independent samples was used to compare the quantitative variables between groups. Categorical variables are presented as frequencies and percentages and compared using the chi-squared test with Yate’s correction. The correlations between the variables were examined by determining Pearson’s coefficient. A *p*-value of <0.05 was considered to be statistically significant. The ability of the LUS score to predict 30-day mortality was assessed by measuring the area under the receiver operating characteristics curve (ROC) (AUC). The best threshold of the ROC curve was chosen using bootstrap analysis and the maximization of the Youden index.

All data were collected and entered into an Excel database (Microsoft Office 2016), and statistical analyses were performed using SPSS (IBM SPSS Statistics 25 Version, Inc., Chicago, IL, USA).

## 4. Results

A total of 1826 patients were examined, and their baseline characteristics are summarized in Figure 2. Based on the nasal swab result, patients were divided into COVID-19-positive (*n* = 617) and negative (*n* = 843) groups, the characteristics of which are shown in the Appendix A. In 312 cases, the nasal swab was not performed; in 11 cases, the swab had poor cellularity, and in 43 cases, the test results were not available. Table 1 shows the main blood gas analysis differentiated by the COVID-19 nasal swab result.

COVID-19-positive patients came to the emergency department at 5.0 ± 6.9 days from the onset of symptoms.

The analysis of COVID-19-positive patients in consideration of the outcome from the emergency department showed that 30% were discharged, 60.8% were hospitalized in an ordinary ward, 6% were hospitalized in sub-intensive care, 0.5% were hospitalized in intensive care, 0.3% died, and 1.6% were transferred to another institution. The analysis performed on the intensity of care during the hospitalization showed that 40.6% of the COVID-19 patients were discharged, 43.1% were hospitalized in low-intensity care ward, 6.1% were hospitalized in the sub-intensive care unit, and 5.1% were hospitalized in the intensive care unit. Regarding oxygen support provided to patients during hospitalization, 16.7% of COVID-19-positive patients required conventional oxygen therapy, 10.5% required support through Ventimask, 8.6% required oxygen therapy by a facial mask and reservoir, 6.5% required continuous positive airway pressure (CPAP), 0.5% required non-invasive ventilation (NIV), 0.8% required high-flow nasal cannula (HFNC), and 4.4% required orotracheal intubation and invasive ventilation. Regarding whole intra-hospital mortality, COVID-19-positive and -negative patients had 15% and 8.7% mortality rates, respectively (*p* < 0.001). The 30-day mortality of COVID-19 patients was 17.9%.

### Lung Ultrasound Data

A total of 646 lung ultrasound exams were performed (193 in patients with a positive swab and 453 in patients with a negative swab). Ultrasound showed signs of interstitial disease (B lines) in 72.5% of examined COVID-19-positive cases. An irregularity of the pleural line was reported in 31.8% of COVID-19 patients, consolidations were highlighted in 30.5% of COVID-19 patients, and pleural effusion was observed in 8% of COVID-19 patients. A comparison with the ultrasound data performed for COVID-19-negative patients showed a statistically significant difference for all the analyzed signs (Table 2). The analysis of the mean LUS scores showed a statistically significant difference in the group of patients with a positive swab (3.6 ± 4.8) from those with a negative swab (1.8 ± 3.6) (*p* < 0.001) (Table 3). 

In patients with a positive nasal swab, the LUS score was inversely correlated with the P/F ratio (*p* < 0.001; r = −0.56) and the S/F ratio (*p* < 0.001; r = −0.42), and it was directly correlated with the value of delta (A–a) (*p* < 0.001; r = 0.58), the value of the increase in delta (A–a) with respect to the expected (*p* < 0.001; r = 0.53), and the respiratory rate (*p* < 0.001; r = 0.35); the LUS score did not correlate with the pCO_2_ value (*p* = 0.06; r = −0.17) (Figure 3, Figure 4 and Figure 5).

Furthermore, LUS scores were found to increase with increasing levels of the intensity of care, passing from a mean value of 1.4 in discharged patients to 8.6 in patients who were hospitalized in an intensive care unit (ICU) (see Appendix A). Patients who died in the emergency department presented a mean LUS score of 19 ± 9.8. Furthermore, the LUS scores increased in agreement with the level of oxygen support provided, ranging from a mean value of 1.8 in patients who did not require oxygen to a value of 10.0 in patients who required a orotracheal intubation (OTI) (see Appendix A). Considering the group of COVID-19-positive patients, subdivided into patients who survived and patients who died at 30 days, there was a statistically significant difference in the LUS scores (3.0 ± 4.1 vs. 11.3 ± 8.4; *p* < 0.001) (see Appendix A) (Table 4). The analysis of the ROC curves against 30-day mortality in COVID-19 patients showed AUC values of 0.816 for the LUS score, 0.054 for the P/F ratio, 0.777 for the delta (A–a) increase, and 0.825 for the delta (A–a) value. The Youden index calculated for the LUS score curve showed that a value of 7.5 presented a sensitivity of 83% and specificity of 89% (Figure 6) (see Appendix A).

## 5. Discussion

Our work demonstrates that some clinical features are mostly detected in patients with a positive nasal swab. Fever, cough, and ageusia were found to be the most specific symptoms of patients with a positive swab.

The blood gas analysis data showed an alteration of gas exchange in COVID-19 patients, with lower values of the P/F ratio, S/F ratio, and ROX index and higher values of delta (A–a). Those data are in agreement with the results of some previous published papers on COVID-19 patients [13,14].

When performing quantitative ultrasound analyses of lung damage, B-lines were mostly detected (72.5% of cases), followed by the thickening of the pleural line, consolidations, and pleural effusions. A comparison of COVID-19 patients to other patients illustrated that all ultrasound signs showed statistically significant differences, with a greater presence of signs in COVID-19-positive subjects. This finding is in agreement with previous works [4,5,7,15,16].

Notably, the detection of those signs could be influenced by the clinical phase of COVID-19 infection; indeed, lung consolidation, a sign of the total and complete alteration of the lung parenchyma, is more typical of advanced stages of the disease, while B lines and pleural line changes seem to be early signs [17,18]. In our work, patients were mainly examined during early stages of the disease (average of five days from the onset of symptoms), thus justifying the fact that the B lines were the most frequent ultrasound sign.

According to the literature, pleural effusion seems to not be a specific sign of COVID-19 pneumonia, although in our work, there was a mild prevalence in the COVID-19 group [19].

The LUS score scheme has been employed in order to provide a quantitative ultrasound assessment. Our analysis and comparison of the means of the LUS scores showed that patients with a positive swab had higher values than those with a negative swap (3.6 vs. 1.8, respectively).

The LUS score was inversely correlated with the P/F and S/F ratios and directly correlated with the delta (A–a) and the delta increase. Therefore, the LUS score was found to be correlated to the alteration of gas exchange and thus to pathological changes affecting the lung parenchyma. These data are in agreement with the recent work of Secco et al., who reported similar correlations [20].

In our work, we demonstrated that LUS scores increased with the provided level of oxygen support; COVID-19 patients requiring orotracheal intubation and mechanical ventilation presented a mean LUS score of 10.0, and patients treated with mask and reservoir had a mean LUS score of 16 (these patients were often suffering from severe respiratory failure without option for intensive treatment due to age and/or comorbidities).

Seiler et al. reported an LUS score of 19.5 as a cut-off for requirement for invasive mechanical ventilation [21].

An analysis of the LUS scores of the patients according to the type of hospitalization ward showed that there was a corresponding increase in the ultrasound score as the intensity of care increased. Therefore, an LUS score of >9 was related to patients with sub-intensive/ICU hospitalization.

A recent work by Persona et al. demonstrated a median LUS score of 27.5 in COVID-19 patients with acute respiratory failure at ICU admission [22].

Another work by Tombini et al. showed that an LUS score of >20 presented the best diagnostic accuracy for the primary outcome (endotracheal intubation, no active further management or death); in the same work, an LUS score of < 10 presented the best diagnostic accuracy for the secondary outcome (discharge from the emergency room) [23].

Rubio-Gracia and colleagues performed lung ultrasounds during the first 72 h after admission, and they demonstrated that an LUS score of >22 presented the best diagnostic accuracy for in-hospital death and/or admission to the intensive care unit [24].

COVID-19 infection was found to be characterized by higher mortality rate than a non-COVID-19 population (15% vs. 8.7%, respectively). This higher mortality rate was also confirmed 30 days after admission to the emergency room. COVID-19-positive patients who died had a higher mean LUS score (11.3) than those who survived (3.0). An LUS score of greater than 19 detected in the emergency department was found to be related to an higher risk of early death. 

The analysis of the ROC curves for 30-day mortality in COVID-19 patients showed an AUC of 0.816 for the LUS score, with a score of >7.5 presenting a sensitivity of 83% and specificity of 89%. A recent work by Secco et al. showed that an LUS score of >13 had a 77.2% sensitivity and a 71.5% specificity (AUC = 0.814; *p* < 0.001) in predicting mortality [20]. In our work, we also compared the ROC curve of the delta (A–a) values found in the blood gas analysis, and we found an AUC comparable to that of the LUS score (AUC = 0.825). These data may suggest that the two methods have a significant diagnostic capacity, which can be further improved if performed together; on the other hand, lung ultrasound alone can provide important information for the clinical management of patients in out-of-hospital diagnostic settings.

Several studies have presented different LUS scores as predictors of mortality and hospitalization in intensive care [20,21,22,23,24,25]; those differences were probably due to the different care settings of the various hospitals. In our case, many of the examined patients went to the emergency room few days after the onset of symptoms (mean of 5 days). This made it possible to immediately evaluate patients (often during a non-critical phase of the disease), allow for the best therapy to be immediately set in progress, and establish the best care setting.

Therefore, the LUS score determined at the time of the first evaluation seems to be a potential predictor of mortality or clinical worsening in COVID-19 patients. 

## 6. Study Limitations

Our work had some limitations. First, the retrospective nature of the study affected the numerical non-uniformity and baseline characteristics of the two comparison groups. Some data were not available for analysis, as they could not be obtained from the consulted sources. The a priori categorization of patients based on their comorbidities was not performed. On one hand, this was a methodological limitation; on the other hand, it fully depicted the real world experience of those patients and could be used to validate the method in all patients.

We should reinforce the concept that lung ultrasound findings are not specific for COVID-19 pneumonia; indeed, they have been previously described for pulmonary congestion, pulmonary fibrosis, lymphocytic or granulomatous lung interstitial diseases, atelectasis, lymphangitis, lung contusion, cardiac failure, and acute respiratory distress syndrome [26,27]. Moreover, similar lung ultrasound patterns were previously described for interstitial pneumonia caused by chlamydia, pneumocystis, measles, influenza virus A H7N9, and influenza virus H1N1 [28,29]. Furthermore, inter-observer and intra-observer variabilities were not calculated in our work.

## 7. Conclusions

Lung ultrasound allows for an optimal assessment of lung parenchyma damage due to the specific distribution of the disease at the peripheral level [4,5].

Patients who died from COVID-19 were characterized by a mean LUS score of 11 at presentation to the emergency department. An LUS score of >7.5 demonstrated a sensitivity of 83% and specificity of 89% against 30-day mortality in COVID-19 patients. Lung ultrasound seems to be an optimal first level method for pneumonia detection in patients with suspected SARS-CoV-2 infection. Lung ultrasound is an easy-to-perform diagnostic tool that provides relevant clinical data, with no contraindications and side effects.

## Figures and Tables

**Figure 1 jcm-11-02067-f001:**
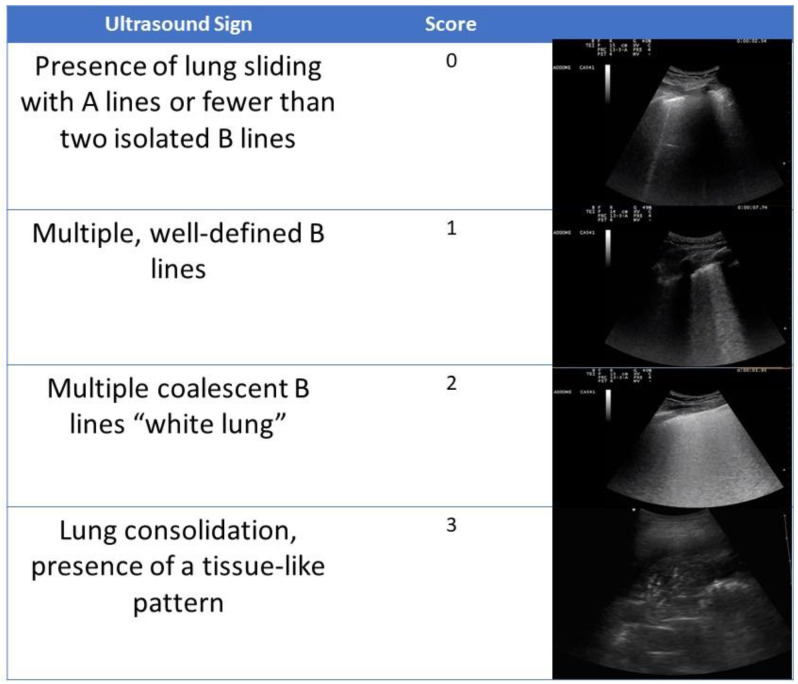
The sum of the number of the 12 examined lung fields provided a final result (LUS score); the LUS score therefore ranged from 0 to 36. The patient was preferentially examined in the sitting position; in case of forced supine position, posterior scans were performed by rolling the patient on their side.

**Figure 2 jcm-11-02067-f002:**
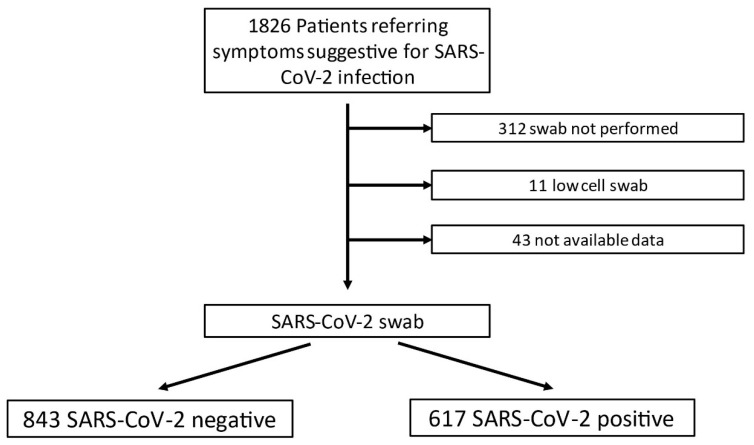
Flowchart scheme with number of patients included and excluded in the study.

**Figure 3 jcm-11-02067-f003:**
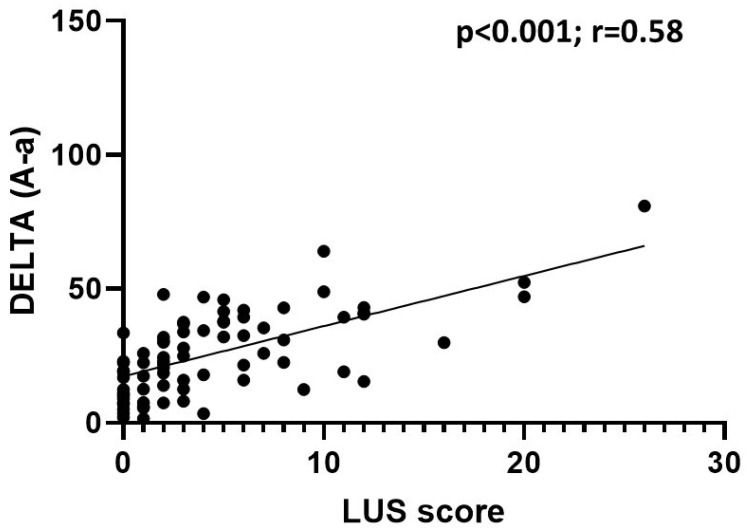
Correlation between LUS score and delta (A–a) in COVID-19 patients. The data show a statistically significant direct correlation.

**Figure 4 jcm-11-02067-f004:**
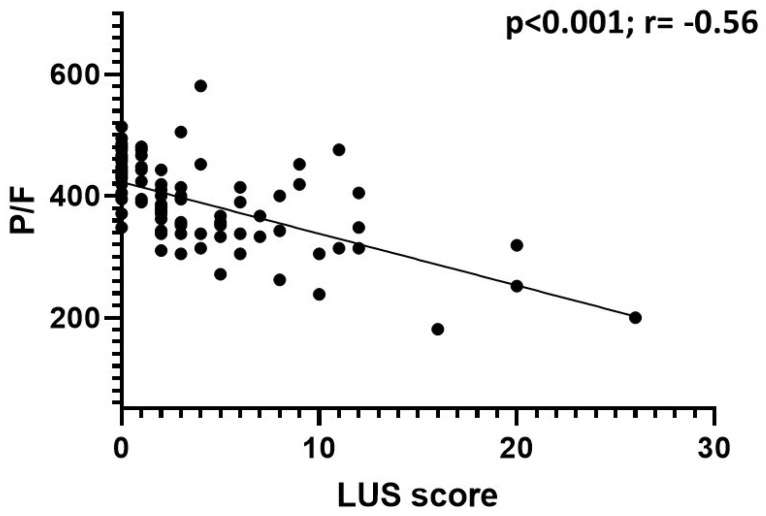
Correlation between LUS score and P/F ratio in COVID-19 patients. The data show a statistically significant inverse correlation.

**Figure 5 jcm-11-02067-f005:**
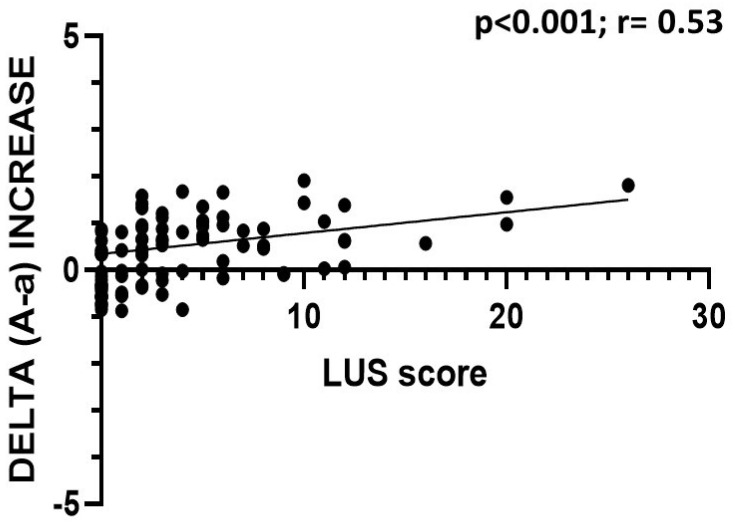
Correlation between LUS score and delta (A-a) increase in COVID-19 patients. The data show a statistically significant direct correlation.

**Figure 6 jcm-11-02067-f006:**
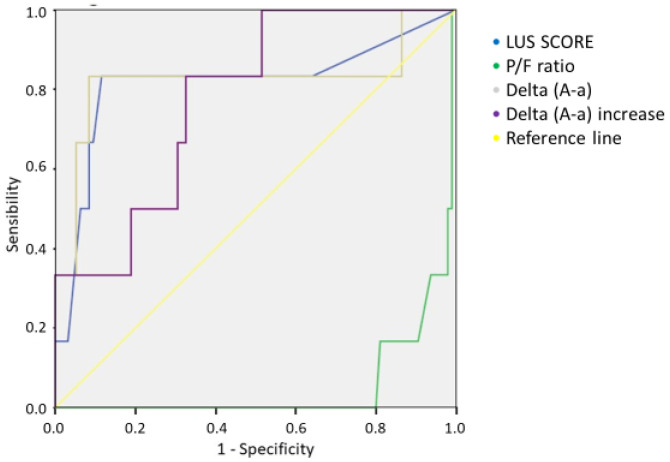
Comparison of ROC curves of LUS score, P/F ratio, delta (A–a), and delta (A–a) increase for the 30-day mortality of COVID-19 patients.

**Table 1 jcm-11-02067-t001:** Respiratory and blood gas analytical data found in the emergency department. S/F: SpO_2_/FiO_2_; P/F: pO_2_/FiO_2_; ROX index: SpO_2_/FiO_2_ to respiratory rate; DELTA (A–a): oxygen alveolar–arterial gradient. Continuous variables are expressed as mean ± standard deviation.

	All Patients (1826)	COVID-19 + (617)	COVID-19 − (843)	*p*-Value
S/F	449.9 ± 59.0	440.8 ± 69.1	453.3 ± 53.3	<0.001
pCO_2_ (mmHg)	35.0 ± 8.5	33.7 ± 6.7	35.9 ± 10.3	<0.001
P/F	386.6 ± 157.8	348.2 ± 104.2	400.0 ± 202.3	<0.001
HCO_3_ (mmol/L)	24.7 ± 3.2	24.4 ± 3.8	24.7 ± 2.6	0.45
LAC (mmol/L)	1.3 ± 1.3	1.4 ± 1.6	1.2 ± 0.9	0.19
DELTA (A–a)	23.4 ± 33.4	32.2 ± 21.7	20.2 ± 42.6	<0.001
DELTA (A–a) expected	18.0 ± 5.5	19.5 ± 4.8	18.3 ± 5.4	<0.001
DELTA (A–a) increase	0.29 ± 2.0	0.6 ± 1.1	0.0 ± 2.8	<0.001
ROX index	24.9 ± 7.7	23.7 ± 7.6	25.2 ± 7.6	<0.001

**Table 2 jcm-11-02067-t002:** Findings of the four main pathological lung ultrasound signs and comparison between the two groups; categorical variables are presented as frequencies and percentages, and they were compared using the chi-squared test with Yate’s correction.

Lung Ultrasound Signs	COVID-19 + (193)	COVID-19 − (453)	*p*-Value
B LINES > 3/scan area	140 (72.5%)	210 (46.3%)	<0.001
Pleural Line Irregularity	61 (31.8%)	98 (21.6%)	<0.001
Consolidation	59 (30.5%)	94 (20.7%)	<0.001
Pleural Effusion	16 (8.2%)	32 (7.0%)	0.040

**Table 3 jcm-11-02067-t003:** LUS scores differentiated by SARS-CoV-2 nasal swab result; continuous variables are expressed as mean ± standard deviation.

	All Patients (646)	COVID-19 + (193)	COVID-19 − (453)	*p*-Value
LUS score	2.0 ± 3.7	3.6 ± 4.8	1.8 ± 3.6	<0.001

**Table 4 jcm-11-02067-t004:** Comparison of blood gas analysis and ultrasound data between dead and surviving COVID-19 patients.

	Dead	Surviving	*p*-Value
pCO_2_	32.5 ± 6.8	34.0 ± 6.6	0.046
P/F	255.7 ± 85.7	369.5 ± 94.8	<0.001
DELTA (A–a)	47.8 ± 19.3	28.8 ± 19.4	0.018
LUS Score	11.3 ± 8.4	3.0 ± 4.1	<0.001

## Data Availability

Data are available on request from the corresponding author.

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
