# Peer review of "Role of Lung Ultrasound in the Management of Patients with Suspected SARS-CoV-2 Infection in the Emergency Department"

_jcm, 2022, doi:10.3390/jcm11082067_

Round 1
Reviewer 1 Report
In emergency departments, the use of lung ultrasound (LUS) as a primary survey tool in patients clinically suspected of having COVID-19 infection provides an immediate evaluation of the lung.
This article is similar in design and aim to a number of other articles that have been published in the last 1-2 years.
The writing is clear and easily understandable. There are some minor grammatical errors or typos. The writing may benefit a review by a professional writer.
Abstract. Acronyms may be defined when used for the first time (e.g. LUS) and used anytime it appears.
Abstract. P/F ratio or Delta (A-a) terms should be explained in order to help the readers.
Abstract. I would change the following sentence “ Lung ultrasound seems to be an optimal first level method for pneumonia detection in patients with suspected SARS-CoV-2 infection” in “LUS seems to be an optimal first level method for pneumonia detection and risk stratification in patients with suspected SARS-CoV-2 infection.”
Abstract. Authors claim that “The aim of our study was to evaluate the use of lung ultrasound as a diagnostic tool for diagnosing SARS-CoV-2 pneumonia”. However, it’s not clear whether all recruited patients (both COVID-19 positive and negative) were admitted to the ED with clinical signs/symptoms of pneumonia. Looking at Table 3 seems only few patients were suspected for pneumonia.
I would particularly interested in some sub-analysis:
-- May LUS also help in early diagnosis? Authors should consider only those patients with mild COVID-19+ disease (olygosymptomatic or asymptomatic patients, or those patients without respiratory symptoms and fever) and in the first few days after symptoms onset. Even in this specific subset of patients, LUS was a sensitive method to early diagnose COVID-19 and its risk stratification was usefull to predict final outcome?
-- For a patient with respiratory failure, a normal LUS score could rule out COVID-19 pneumonia and orient the examiner toward different diagnosis. Did you confirm this aspect?
Table 1 should be removed. A flow-chart scheme with number of patients included and excluded is preferred.
In particular, how many patients were excluded due to underlying comorbidity characterized by B-lines?
In fact, several disease can present with B-lines, regardless of COVID-19 severity, such as fluid overload, renal failure, pulmonary fibrosis, etc (Allinovi et al, 2020). Mention a few differential diagnosis.
Table 1-7. There are several errors in p-value calculation. Did you use Chi-square? In particular: COPD 10.7 % vs 10.5% with a p-value <0.0001 Of course this is not correct, but please, consider all the others values because they seems incorrect.
Table 6. What’s the meaning of “B lines” ? Presence or absence?
Authors should include a threshold to identify a “normal value” of B lines at LUS score. Consequently, in table 6, should appear two lines on B lines, one with B line number < … and one with B line number > …
The quality of images 5-8 should be increased.
More information should be mentioned in the legends of Figure 2-8. Authors should be more descriptive about the meaning of these figures.
In general, there are too many figures and tables. Consequently, Authors could consider to (1) remove some of them, (2) put some of them in supplementary material, or (3) unify 2 or more into one.
LUS findings for COVID-19 pneumonia are not specific and have been previously described for pulmonary congestion, pulmonary fibrosis, flogistic or granulomatous lung interstitial diseases, atelectasis, lymphangitis, lung contusion, cardiac failure and acute respiratory distress syndrome (Dietrich et al, 2016; Allinovi et al, 2020). A LUS pattern similar to COVID-19 was previously described for interstitial pneumonia caused by Chlamydia, Pneumocystis, measles, influenza virus A H7N9 and influenza virus H1N1 (Allinovi et al, 2020; Lo Giudice et al, 2008). Even this concept should be included.
The following references should be included:
-- Dietrich CF, Mathis G, Blaivas M, Volpicelli G, Seibel A, Wastl D, Atkinson NS, Cui XW, Fan M, Yi D. Lung B-line artefacts and their use. J Thorac Dis. 2016 Jun;8(6):1356-65.
-- Allinovi M, Parise A, Giacalone M, Amerio A, Delsante M, Odone A, Franci A, Gigliotti F, Amadasi S, Delmonte D, Parri N, Mangia A. Lung Ultrasound May Support Diagnosis and Monitoring of COVID-19 Pneumonia. Ultrasound Med Biol. 2020 Nov;46(11):2908-2917.
-- Lo Giudice V, Bruni A, Corcioni E, Corcioni B. Ultrasound in the evaluation of interstitial pneumonia. J Ultrasound. 2008 Mar; 11(1): 30–38.
-- Poggiali Erika, Vercelli Andrea, Maria Grazia Cillis, Eva Ioannilli, Teresa Iannicelli, Magnacavallo Andrea. Triage decision-making at the time of COVID-19 infection: the Piacenza strategy. Intern Emerg Med. 2020 Aug;15(5):879-882.
Author Response
We thank the reviewer for his attention and comments. We provide a point-by-point answer below
In emergency departments, the use of lung ultrasound (LUS) as a primary survey tool in patients clinically suspected of having COVID-19 infection provides an immediate evaluation of the lung.
This article is similar in design and aim to a number of other articles that have been published in the last 1-2 years.
The writing is clear and easily understandable. There are some minor grammatical errors or typos. The writing may benefit a review by a professional writer.
R: a grammatical revision has been performed
Abstract. Acronyms may be defined when used for the first time (e.g. LUS) and used anytime it appears.
R: we have specified acronyms
Abstract. P/F ratio or Delta (A-a) terms should be explained in order to help the readers.
R: we have fully described those terms
Abstract. I would change the following sentence “ Lung ultrasound seems to be an optimal first level method for pneumonia detection in patients with suspected SARS-CoV-2 infection” in “LUS seems to be an optimal first level method for pneumonia detection and risk stratification in patients with suspected SARS-CoV-2 infection.”
R: amended as requested
Abstract. Authors claim that “The aim of our study was to evaluate the use of lung ultrasound as a diagnostic tool for diagnosing SARS-CoV-2 pneumonia”. However, it’s not clear whether all recruited patients (both COVID-19 positive and negative) were admitted to the ED with clinical signs/symptoms of pneumonia. Looking at Table 3 seems only few patients were suspected for pneumonia.
R: we stated: “all patients who were referred for symptoms suspected for SARS-CoV-2 infection”, therefore all the patients referring one of those symptoms (although non-specific) reported in table 3, were evaluated as described
I would particularly interested in some sub-analysis:
-- May LUS also help in early diagnosis? Authors should consider only those patients with mild COVID-19+ disease (olygosymptomatic or asymptomatic patients, or those patients without respiratory symptoms and fever) and in the first few days after symptoms onset. Even in this specific subset of patients, LUS was a sensitive method to early diagnose COVID-19 and its risk stratification was usefull to predict final outcome?
R: most of our patients presented with mild symptoms (mean P / F ratio: 348), and presented early in the emergency room with an average of 5 days from the onset of symptoms. Therefore the data of our work seem to be valid especially in this category of patients.
-- For a patient with respiratory failure, a normal LUS score could rule out COVID-19 pneumonia and orient the examiner toward different diagnosis. Did you confirm this aspect?
R: unfortunately, in the study design we did not divide patients into categories regarding comorbidities or degree of respiratory failure. That limits the possibility to analyze those data. In addition, sars-cov-2 positive patients without pneumonia should be differentiated from those infected with pneumonia.
In a subanalysis of the work (data not included) in a series of COVID-19 positive patients, who had performed both ultrasound and CT (gold standard comparison method) only one case out of 32 presented a discrepancy between the two methods; moreover, a LUS score with a value of 1.5 presented an 83% sensitivity and 100% specificity towards the CT finding of at least 1 area of ​​ground glass.
Table 1 should be removed. A flow-chart scheme with number of patients included and excluded is preferred.
R: amended as requested
In particular, how many patients were excluded due to underlying comorbidity characterized by B-lines?
In fact, several disease can present with B-lines, regardless of COVID-19 severity, such as fluid overload, renal failure, pulmonary fibrosis, etc (Allinovi et al, 2020). Mention a few differential diagnosis.
R: only patients who had not performed and it was not possible to obtain the result of the swab were excluded from the analysis. Patients who were not diagnosed with acute sars-cov-2 infection were included in the negative patient group. As previously mentioned, a categorization of patients based on their comorbidities was not performed. While this is a methodological limitation, on the other it fully depicts the daily reality of patients, and I validate the use of the method in all patients. A sentence was included in the limits section of the study.
Table 1-7. There are several errors in p-value calculation. Did you use Chi-square? In particular: COPD 10.7 % vs 10.5% with a p-value <0.0001 Of course this is not correct, but please, consider all the others values because they seems incorrect.
R: we revised the results
Table 6. What’s the meaning of “B lines” ? Presence or absence?
R: we specified the data
Authors should include a threshold to identify a “normal value” of B lines at LUS score. Consequently, in table 6, should appear two lines on B lines, one with B line number < … and one with B line number > …
R: we specified the data
The quality of images 5-8 should be increased.
R: we have excluded some figures and tables
More information should be mentioned in the legends of Figure 2-8. Authors should be more descriptive about the meaning of these figures.
In general, there are too many figures and tables. Consequently, Authors could consider to (1) remove some of them, (2) put some of them in supplementary material, or (3) unify 2 or more into one.
R: we have excluded some figures and tables
LUS findings for COVID-19 pneumonia are not specific and have been previously described for pulmonary congestion, pulmonary fibrosis, flogistic or granulomatous lung interstitial diseases, atelectasis, lymphangitis, lung contusion, cardiac failure and acute respiratory distress syndrome (Dietrich et al, 2016; Allinovi et al, 2020). A LUS pattern similar to COVID-19 was previously described for interstitial pneumonia caused by Chlamydia, Pneumocystis, measles, influenza virus A H7N9 and influenza virus H1N1 (Allinovi et al, 2020; Lo Giudice et al, 2008). Even this concept should be included.
R: we have included a sentence in the section limitation
The following references should be included:
-- Dietrich CF, Mathis G, Blaivas M, Volpicelli G, Seibel A, Wastl D, Atkinson NS, Cui XW, Fan M, Yi D. Lung B-line artefacts and their use. J Thorac Dis. 2016 Jun;8(6):1356-65.
-- Allinovi M, Parise A, Giacalone M, Amerio A, Delsante M, Odone A, Franci A, Gigliotti F, Amadasi S, Delmonte D, Parri N, Mangia A. Lung Ultrasound May Support Diagnosis and Monitoring of COVID-19 Pneumonia. Ultrasound Med Biol. 2020 Nov;46(11):2908-2917.
-- Lo Giudice V, Bruni A, Corcioni E, Corcioni B. Ultrasound in the evaluation of interstitial pneumonia. J Ultrasound. 2008 Mar; 11(1): 30–38.
-- Poggiali Erika, Vercelli Andrea, Maria Grazia Cillis, Eva Ioannilli, Teresa Iannicelli, Magnacavallo Andrea. Triage decision-making at the time of COVID-19 infection: the Piacenza strategy. Intern Emerg Med. 2020 Aug;15(5):879-882.
R: all references have been included

Reviewer 2 Report
This retrospective cohort study of patients presenting to the ER of a tertiary hospital in Italy presenting COVID-19 related symptoms explored the role of Lung Ultrasound for diagnosing SARS COV-2. Most importantly, the authors examined the prognostic significance of high Lung Ultrasound score on 30-day mortality.
The article is well structured providing tables and figures when needed without repeating data previously reported in results section. Also the authors present a nice figure explaining how lung ultrasound examination is performed in every day clinical practice.
The study is meticulously performed in terms of the thorough statistical analyses.
In my opinion, the results of this study are of extreme interest because they contain appropriate findings according to current knowledge and provide additional information concerning the importance of lung ultrasound examination in COVID-19 patients. Most importantly, they highlight that a LUS score value > 7.5 displays a sensitivity of 83% and specificity of 89% for 30-days mortality in COVID-19 patients which is extremely interesting.
Finally, authors provide clear take-home messages and the study limitations are clearly highlighted.
Major Points
1) The only part of the article that seems problematic is the Conclusions section. In my opinion, it is extended and repeats some knowledge presented in the introduction. I believe, that << Ultrasound also correlates... of care intensity >> is not needed, since it does not add further significance to the presented findings.
2) In the limitations section you should mention that concerning the lung ultrasound examination, the inter-observer and intra-observer variability was not calculated.
Minor Points
1) In the abstract section << A retrospective analysis was performed...infection>> .Please report the number of patients that were included in the study.
2) Abstract: << Covid-19 died patients>>. Please change to << Patients who died from Covid-19 had a mean...department>>.
3) Abstract: << A LUS score... against 30-days mortality in COVID-19 patients>>. You should change to: << for 30-day mortality in covid-19 patients>>.
4) Section 2.1 Please correct the typo
5) Please correct p=0.000 that is presented throughout the tables of the manuscript to p<0.001
6) Table 5. Please give the full definitions of S/F, P/F (Delta A-a), ROX index abbreviations
7) Results << The analysis of the maximal type of care intensity during the all hospitalization >> Please edit this sentence
8) Results << patients throughout the all hospitalization>> Please correct accordingly
9) <<subdivided by survived and died patients>>. Please change to <<subdivided into survived and died patients>>.
10) I think that the captions of Figures 5 and 6 are presented contrarily
11) Discussion << had a higher values than the negative ones (3.6 vs 1.8)>> Please delete <<a>>
12) << The dead patients with positive swab had a higher LUS score mean (11.3) than survived (3.0)>> . You should edit this sentence
13) << of the ROC curves against 30-days mortality>> See comment 3
14) << Therefore, the LUS score, performed at the time of the first evaluation seems to be a potentially predictor of mortality or clinical worsening in COVID-19 patients.>> Please change to << a potential predictor >>
Author Response
This retrospective cohort study of patients presenting to the ER of a tertiary hospital in Italy presenting COVID-19 related symptoms explored the role of Lung Ultrasound for diagnosing SARS COV-2. Most importantly, the authors examined the prognostic significance of high Lung Ultrasound score on 30-day mortality.
The article is well structured providing tables and figures when needed without repeating data previously reported in results section. Also the authors present a nice figure explaining how lung ultrasound examination is performed in every day clinical practice.
The study is meticulously performed in terms of the thorough statistical analyses.
In my opinion, the results of this study are of extreme interest because they contain appropriate findings according to current knowledge and provide additional information concerning the importance of lung ultrasound examination in COVID-19 patients. Most importantly, they highlight that a LUS score value > 7.5 displays a sensitivity of 83% and specificity of 89% for 30-days mortality in COVID-19 patients which is extremely interesting.
Finally, authors provide clear take-home messages and the study limitations are clearly highlighted.
We thank the reviewer for his attention and comments. We provide a point-by-point answer below
Major Points
- The only part of the article that seems problematic is the Conclusions section. In my opinion, it is extended and repeats some knowledge presented in the introduction. I believe, that << Ultrasound also correlates... of care intensity >> is not needed, since it does not add further significance to the presented findings.
R: we delete the sentence
- In the limitations section you should mention that concerning the lung ultrasound examination, the inter-observer and intra-observer variability was not calculated.
R: we included a sentence in the section limitation
Minor Points
- In the abstract section << A retrospective analysis was performed...infection>> .Please report the number of patients that were included in the study.
R: amended as requested
- Abstract: << Covid-19 died patients>>. Please change to << Patients who died from Covid-19 had a mean...department>>.
R: amended as requested
- Abstract: << A LUS score... against 30-days mortality in COVID-19 patients>>. You should change to: << for 30-day mortality in covid-19 patients>>.
R: amended as requested
- Section 2.1 Please correct the typo
R: amended as requested
- Please correct p=0.000 that is presented throughout the tables of the manuscript to p<0.001
R: amended as requested
- Table 5. Please give the full definitions of S/F, P/F (Delta A-a), ROX index abbreviations
R: we specified all the abbreviations
- Results << The analysis of the maximal type of care intensity during the all hospitalization >> Please edit this sentence
R: we changed the sentence
- Results << patients throughout the all hospitalization>> Please correct accordingly
R: we changed the sentence
- <<subdivided by survived and died patients>>. Please change to <<subdivided into survived and died patients>>.
R: amended as requested
10) I think that the captions of Figures 5 and 6 are presented contrarily
R: we corrected the captions
- Discussion << had a higher values than the negative ones (3.6 vs 1.8)>> Please delete <<a>>
R: amended as requested
- << The dead patients with positive swab had a higher LUS score mean (11.3) than survived (3.0)>> . You should edit this sentence
R: we edited the sentence
- << of the ROC curves against 30-days mortality>> See comment 3
R: amended as requested
- << Therefore, the LUS score, performed at the time of the first evaluation seems to be a potentially predictor of mortality or clinical worsening in COVID-19 patients.>> Please change to << a potential predictor >>
R: amended as requested

Round 2
Reviewer 1 Report
The authors have made useful changes in the paper and incorporated all of my suggestions.